# Practical approaches in evaluating validation and biases of machine learning applied to mobile health studies
Johannes Allgaier ⓘ ✉ & Rüdiger Pryss ⓘ

## Abstract

**Background** Machine learning (ML) models are evaluated in a test set to estimate model performance after deployment. The design of the test set is therefore of importance because if the data distribution after deployment differs too much, the model performance decreases. At the same time, the data often contains undetected groups. For example, multiple assessments from one user may constitute a group, which is usually the case in mHealth scenarios.

**Methods** In this work, we evaluate a model's performance using several cross-validation train-test-split approaches, in some cases deliberately ignoring the groups. By sorting the groups (in our case: Users) by time, we additionally simulate a concept drift scenario for better external validity. For this evaluation, we use 7 longitudinal mHealth datasets, all containing Ecological Momentary Assessments (EMA). Further, we compared the model performance with baseline heuristics, questioning the essential utility of a complex ML model.

**Results** Hidden groups in the dataset leads to overestimation of ML performance after deployment. For prediction, a user's last completed questionnaire is a reasonable heuristic for the next response, and potentially outperforms a complex ML model. Because we included 7 studies, low variance appears to be a more fundamental phenomenon of mHealth datasets.

**Conclusions** The way mHealth-based data are generated by EMA leads to questions of user and assessment level and appropriate validation of ML models. Our analysis shows that further research needs to follow to obtain robust ML models. In addition, simple heuristics can be considered as an alternative for ML. Domain experts should be consulted to find potentially hidden groups in the data.

## Plain Language Summary

Computational approaches can be used to analyse health-related data collected using mobile applications from thousands of participants. We tested the impact of some participants being represented multiple times or some not being counted properly within the analysis. In this context, we label a multi-represented participant a group. We find that ignoring such groups can lead to false estimation of health-related predictions. In some cases, simpler quantitative methods can outperform complex computational models. This highlights the importance of monitoring and validating results conducted by complex computational models and confers the use of simpler analytical methods in its place.

When machine learning models are applied to medical data, an important question is whether the model learns subject-specific characteristics (not desired effect) or disease-related characteristics (desired effect) between an input and output. A recent paper by Kunjan et al.[1] describes this very well at the example of classification and EEG disease diagnosis. In the Kunjan et al. paper, this is discussed using different variants of cross-validation. It is well shown that the type of validation can cause extreme differences. Older work has evaluated different cross-validation techniques on datasets with different recommendations for the number of optimal folds[2,3]. We transfer and adapt this idea to mHealth data and the application of machine-learning-based classification and raise new questions about this. To this end, we will briefly explain the background. Using simple, understandable models rather than complex black box models is a clamor of Rudin et. al., which motivates us to evaluate simple heuristics against complex models[4]. The Cross-Industry Standard Process for Data Mining (CRISP-DM) highlights the importance of subject matter experts to get familiar with a dataset[5]. In turn, familiarity with the dataset is necessary to detect hidden

Institute of Clinical Epidemiology and Biometry, Julius-Maximilians-University Würzburg, Josef-Schneider-Straße 2, Würzburg, Germany.
✉e-mail: johannes.allgaier@uni-wuerzburg.de

groups in the dataset. In our mHealth use cases, one app user that fills out more several questionnaires constitutes a group.

We have developed numerous applications in mobile health in recent years (e.g.[6,7]) and the issue of disease-related or subject-specific characteristics is particularly pronounced in these applications. mHealth applications very often use the principles of Patient-reported Outcome Measures (PROMs) or/and Ecological Momentary Assessments (EMAs)[8]. EMAs have the major goal that users record symptoms several times a day over a longer period. As a result, users of an mHealth solution generate longitudinal data with many assessments. Since not all users respond equally frequently in the applications (as shown by many applications that have been in operation for a long time[9]), the result is a very different number of assessments per user. Therefore, the question arises in the application of machine learning, how the actual learning takes place. In learning, should we group the ratings per user so that a user only appears in either the training set or the testing set, which is correct by design. Or, can we accept that a user's ratings appear in both the training and test sets, since users with many ratings have such a high variance in ratings. Finally, individual users may undergo concept drift in the way they answer questions in many assessments over a long period of time. In such a case, the question also arises as to whether it makes sense to use an individual's ratings separately in the training and testing sets.

In this context, we also see another question as relevant that is not given enough attention: What is an appropriate baseline for a machine learning outcome in studies? As mentioned earlier, some mHealth users fill out thousands of assessments, and do so for years. In this case, there may be questions about whether a previous assessment can reliably predict the next one, and the use of machine learning may be poorly targeted.

With respect to the above research questions, we use another component to further promote the results. We selected seven studies from the pool of developed apps that we will use for the analysis of this paper. Since a total of 7 studies are used, a more representative picture should emerge. However, since the studies do not all have the same research goals, classification tasks need to be found per app to make the overall results comparable. The studies also do not all have the same duration. Even though the studies are not always directly comparable, the setting is very promising as the results will show in the end. Before deriving specific research questions against this background, related work and technical background information will be briefly discussed.

This section surveys relevant literature to contextualize our contributions within the broader field of study. Cawley et al. also address the question of how to minimize the error in the estimator of performance in ground truth. Using synthetic data sets, they argue that overfitting a model is as problematic as selection bias in the training data[10]. However, they do not address the phenomenon of groups in the data. Refaeilzadeh et al. give an overview of common cross-validation techniques such as leave-one-out, repeated k-fold, or hold-out validation[11]. They discuss pros and cons of each kind and mention an *underestimated performance variance* for repeated k-fold cross-validation, but they also do not address the problem with (unknown) groups in the dataset[11]. Schratz et. al. focus on spatial autocorrelation and spatial cross-validation rather than on groups and splitting approaches[12]. Spatial cross-validation is sometimes also referred to as block cross-validation13. They observe large performance differences in the use or non-use of spatial cross validation. By random sampling of train and test samples, a train and test sample might be too close to each other on a geographical space, which induces a selection bias and thus an over-optimistic estimate of the generalization error. They then use spatial cross-validation. We would like to briefly differentiate between *space* and *group*. Two samples belong to the same space if they are geographically close to each other[13]. They belong to the same group if a domain expert assigns them to a group. In our work, multiple assessments belonging to one user form a group. Meyer et al. also evaluate using a spatial cross-validation approach, but also add a time dimension using Leave-Time-Out cross-validation where samples belong to one fold if they fall into a specific time range[14]. This leave-time-out approach is like our *time-cut* approach, which will be

introduced in the methods section. Yet, we are not aware of any related approach on mHealth data like the one we are pursuing in this work.

As written at the beginning of the introduction, we want to evaluate how much the model's performance depends on specific users (syn. *subjects, patients, persons*) that are represented several times within our dataset, but with a varying number of assessments per user. From previous work, we already know that so-called power-users with many more assessments than most of the other users have a high impact on the models training procedure[15]. We would further like to investigate whether a simple heuristic can outperform complex ensemble methods. Simple heuristics are interesting because they are easy to understand, have a low maintenance requirement, and have low variance, but also generate high bias.

Technically, across studies (i.e., across the seven studies), we investigate simple heuristics at the user and assessment level and compare them to tree-based non-tuned ML ensembles. Tree-based methods have already been proven in the literature on the specific mHealth data used, that is why we use only tree-based methods. The reason for not tuning these models is that we want to be more comparable across the used studies. With these levels of consideration, we would like to elaborate on the following research two questions: First, what is the variance in performance when using different splitting methods for train and test set of mHealth data (RQ1)? Second, in which cases is the development, deployment and maintenance of a ML model compared to a simple baseline heuristic worthwhile when being used on mHealth data?

The present work compares the performance of a tree-based ensemble method if the split of the data happens on two different levels: User and assessment. It further compares this performance to non-ML approaches that uses simple heuristics to also predict the target on a user- or assessment level. To summarize the major findings: First, ignoring users in datasets during cross-validation leads to an overestimation of the model's performance and robustness. Second, for some use cases, simple heuristics are as good as complicated tree-based ensemble methods. Within this domain, heuristics are more advantageous if they are trained or applied at the user level. ML models also work at the assessment level. And third, sorting users can simulate concept drift in training if the time span of data collection is large enough. The results in the test set change due to the shuffling of users.

## Methods

In this section, we first describe how Ecological Momentary Assessments work and how they differentiate from assessments that are collected within a clinical environment. Second, we present the studies and ML use cases for each dataset. Next, we introduce the non-ML baseline heuristics and explain the ML preprocessing steps. Finally, we describe existing train-test-split approaches (cross-validation) and the splitting approaches at the user- and assessment levels.

### Ecological momentary assessments

Within this context, *ecological* means "within the subject's natural environment", and *momentary* "within this moment" and ideally, in real time[16]. Assessments collected in research or clinical environments may cause recall bias of the subject's answers and are not primarily designed to track changes in mood or behavior longitudinally. Ecological Momentary Assessments (EMA) thus increase validity and decrease recall bias. They are suitable for asking users in their daily environment about their state of being, which can change over time, by random or interval time sampling. Combining EMAs and mobile crowdsensing sensor measurements allows for multimodal analyses, which can gain new insights in, e.g., chronic diseases[8,15]. The datasets used within this work have EMA in common and are described in the following subsection.

### The ML use cases

From ongoing projects of our team, we are constantly collecting mHealth data as well as Ecological Momentary Assessments[6,17–19]. To investigate how the machine learning performance varies based on the splits, we wanted

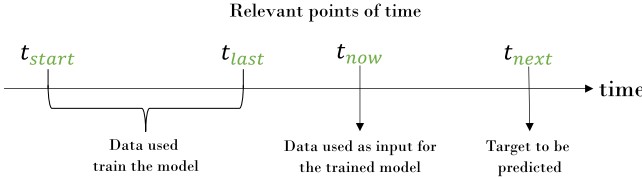

**Fig. 1 | Schematic representation of the relevant four points in time for the understanding of the pipeline.** At time $t_{start}$, the first assessment is given; $t_{last}$ is the last known assessment used for training, whereas $t_{now}$ is the currently available assessment as input for the classifier and the target is predicted at time $t_{text}$.

different datasets with different use cases. However, to increase comparability between the use cases, we created multi-class classification tasks.

We train each model using historical assessments, the oldest assessment was collected at time $t_{start}$, the latest historical assessment at time $t_{last}$. A current assessment is created and collected at time $t_{now}$, a future assessment at time $t_{next}$. Depending on the study design, the actual point of time $t_{next}$ may be in some hours or in a few weeks from $t_{now}$. For each dataset and for each user, we want to predict a feature (synonym, a question of an assessment) at time $t_{next}$ using the features at time $t_{now}$. This feature at time $t_{next}$ is then called the target. For each use case, a model is trained using data between $t_{start}$ and $t_{last}$, and given the input data from $t_{now}$, it predicts the target at time $t_{next}$. Figure 1 gives a schematic representation of the relevant points of time $t_{start}$, $t_{last}$, $t_{now}$, and $t_{next}$.

To increase comparability between the approaches, we used the same model architecture with the same pseudo-random initialisation. The model is a Random Forest classifier with 100 trees and the Gini impurity as the splitting criterion. The whole coding was in Python 3.9, using mostly *scikit-learn*, *pandas* and Jupyter Notebooks. Details can be found on GitHub in the supplementary material.

**The included apps and studies in more detail.** For all datasets that we used in this study, we have ethical approvals (UNITI No. 20-1936-101, TYT No. 15-101-0204, Corona Check No. 71/20-me, and Corona Health No. 130/20-me). The following section provides an overview of the studies, the available datasets with characteristics, and then describes each use case in more detail. An brief overview is given in Table 1 with baseline statistics for each dataset in Table 2.

To provide some more background info about the studies: The analyses happen with all apps on the so-called EMA questionnaires (synonym: assessment), i.e., the questionnaires that are filled out multiple times in all apps and the respective studies. This can happen several times a day (e.g., for the tinnitus study TrackYourTinnitus (TYT)) or at weekly intervals (e.g., studies in the Corona Health (CH) app). Nevertheless, the analysis happens on the recurring questionnaires, which collect symptoms over time and in the real environment through unforeseen (i.e., random) notifications.

The TrackYourTinnitus (TYT) dataset has the most filled-out assessments with more than 110,000 questionnaires as by 2022-10-24. The Corona Check (CC) study has the most users. This is because each time an assessment is filled out, a new user can optionally be created. Notably, this app has the largest ratio of non-German users and the youngest user group with the largest standard deviation. The Corona Health (CH) app with its studies *Mental health for adults, adolescents and physical health for adults* has the highest proportion of German users because it was developed in collaboration with the Robert Koch Institute and was primarily promoted in Germany. Unification of treatments and Interventions for Tinnitus patients (UNITI) is a European Union-wide project, which overall aim is to deliver a predictive computational model based on existing and longitudinal data[19]. The dataset from the UNITI randomized controlled trial is described by Simoes et al.[20].

**TrackYourTinnitus (TYT).** With this app, it is possible to record the individual fluctuations in tinnitus perception. With the help of a mobile device, users can systematically measure the fluctuations of their tinnitus. Via the

**Table 1 | Overview of the mobile applications and the studies involved**

| App Name Logos | TYT | CC | CH | UNITI |
|---|---|---|---|---|
| **Studies Involved & Background** | TYT was launched in 2014 to find out more about daily tinnitus fluctuations and has been a longitudinal observational study using an iOS and Android app. | During the Covid-19 pandemic, CC was developed to provide feedback based on reported symptoms due to scarce Covid testing and knowledge. | CH contains mental and physical health studies for tracking user behavior during the Covid pandemic, targeting adults and adolescents' mental health, adults' physical health, and stress. | UNITI aims to develop a model for patient-specific treatment of tinnitus. |
| **Project Partners** | Tinnitus Research Initiative | Bavarian State Office for Health and Food Safety | Robert Koch Institute | European Union's Horizon 2020 Research and Innovation Programme |

TrackYourTinnitus (TYT)[18], Corona Check (CC)[22], Corona Health (CH)[6], and Unification of Treatments and Interventions for Tinnitus Patients (UNITI)[19] . Headers in bold.

**Table 2 | Baseline statistics and overview of the datasets used**

| Dataset | No. of users | No. of assessments | First assessment from | Dataset span | ∅ Age (Std) | Ratio m/f/d | % rate of GER[2] users |
|---|---|---|---|---|---|---|---|
| TYT | 3303 | 110,983 | 2013-07-18 | 9.20 | 45.0 (14.4) | 67/33/00 | n. A. |
| CC | 13763 | 89,659 | 2020-04-08 | 2.48 | 32.7 (18.0) | 59/39/01 | 36 |
| CHA | 1474 | 11,081 | 2020-07-21 | 2.19 | 41.2 (13.9) | 54/45/01 | 98 |
| CHP | 953 | 5661 | 2020-07-28 | 2.17 | 41.8 (15.2) | 63/37/00 | 98 |
| CHY | 111 | 630 | 2020-08-08 | 2.14 | 15.2 (1.6) | 51/47/01 | n. A. |
| CHS | 374 | 3845 | 2020-12-19 | 1.78 | 40.7 (13.9) | 65/34/01 | 98 |
| UNITI | 763 | 32,443 | 2021-04-13 | 1.46 | 53.0 (12.7) | 57/43/00 | 54 |

Ratio m/f/d is the sex ratio of male, female and diverse users. The dataset span is given in years. Headers in bold.[2]

[2]GER = German

TYT website or the app, users can also view the progress of their own data and, if necessary, discuss it with their physician.

The ML task at hand is a classification task with target variable *Tinnitus distress* at time $t_{now}$ and the questions from the daily questionnaire as the features of the problem. The target's values range in [0, 1] on a continuous scale. To make it a classification task, we created bins with step size of 0.2 resulting in 5 classes. The features are *perception*, *loudness*, and *stressfulness* of tinnitus, as well as the current *mood*, *arousal* and *stress level* of a user, the *concentration level* while filling out the questionnaire, and *perception of the worst tinnitus symptom*. A detailed description of the features was already done in previous works[21]. Of note, the time delta of two assessments of one user at $t_{next}$ and $t_{now}$ varies between users. Its median value is 11 hours.

Unification of Treatments and Interventions for Tinnitus Patients (UNITI). The overall goal of UNITI is to treat the heterogeneity of tinnitus patients on an individual basis. This requires understanding more about the patient-specific symptoms that are captured by EMA in real time.

The use case we created at UNITI is like that of TYT. The target variable encumbrance, coded as *cumberness*, which was also continuously recorded, was divided into an ordinal scale from 0 to 1 in 5 steps. Features also include momentary assessments of the user during completion, such as *jawbone, loudness, movement, stress, emotion*, and questions about momentary tinnitus. The data was collected using our mobile apps[7]. Here, of note: on average, the median time gap between two assessment is 24 hours for each user.

Corona Check (CC). At the beginning of the COVID-19 pandemic, it was not easy to get initial feedback about an infection, given the lack of knowledge about the novel virus and the absence of widely available tests. To assist all citizens in this regard, we launched the mobile health app Corona Check together with the *Bavarian State Office for Health and Food Safety*[22].

The Corona Check dataset predicts whether a user has a Covid infection based on a list of given symptoms[23]. It was developed in the early pandemic back in 2020 and helped people to get quick estimate for an infection without having an antigen test. The target variable has four classes: First, "suspected coronavirus (COVID-19) case", second, "symptoms, but no known contact with confirmed corona case", third, "contact with confirmed corona case, but currently no symptoms", and last, "neither symptoms nor contact".

The features are a list of Boolean variables, which were known at this time to be typically related with a Covid infection, such as fever, a sore throat, a runny nose, cough, loss of smell, loss of taste, shortness of breath, headache, muscle pain, diarrhea, and general weakness. Depending on the answers given by a user, the application programming interface returned one of the classes. The median time gap of two assessments for the same user is 8 hours on average with a much larger standard deviation of 24.6 days.

Corona Health | Mental health for adults (CHA). The last four use cases are all derived from a bigger Covid-related mHealth project called *Corona*

*Health*[6,24]. The app was developed in collaboration with the Robert Koch-Institute and was primarily promoted in Germany, it includes several studies about the mental or physical health, or the stress level of a user. A user can download the app and then sign up for a study. He or she will then receive a baseline one-time questionnaire, followed by recurring follow-ups with between-study varying time gaps. The follow-up assessment of CHA has a total of 159 questions including a full PHQ9 questionnaire[25]. We then used the nine questions of PHQ9 as features at $t_{now}$ to predict the level of depression for this user for $t_{next}$. Depression levels are ordinally scaled from *None* to *Severe* in a total of 5 classes. The median time gap of two assessments for the same user is 7.5 days. That is, the models predict the future in this time interval.

Corona Health | Mental health for adolescents (CHY). Similar to the adult cohort, the mental health of adolescents during the pandemic and its lockdowns is also captured by our app using EMA.

A lightweight version of the mental health questionnaire for adults was also offered to adolescents. However, this did not include a full PHQ9 questionnaire, so we created a different use case. The target variable to be classified on a 4-level ordinal scale is *perceived dejection* coming from the PHQ instruments, features are a subset of quality of live assessments and PHQ questions, such as concernment, tremor, comfort, leisure quality, lethargy, prostration, and irregular sleep. For this study, the median time gap of two follow up assessments is 7.3 days.

Corona Health | Physical health for adults (CHP). Analogous to the mental health of adults, this study aims to track how the physical health of adults changes during the pandemic period.

Adults had the option to sign up for a study with recurring assessments asking for their physical health. The target variable to be classified asks about the constraints in everyday life that arise due to physical pain at $t_{next}$. The features for this use case include aspects like sport, nutrition, and pain at $t_{now}$. The median time gap of two assessments for the same user is 14.0 days.

Corona Health | Stress (CHS). This additional study within the Corona Health app asks users about their stress level on a weekly basis. Both features and target are assessed on a five-level ordinal scale from *never* to *very often*. The target asks for the ability of stress management, features include the first nine questions of the perceived stress scale instrument[26]. The median time gap of two assessments for the same user on average is 7.0 days.

## Baseline heuristics instead of complex ML models?

We also want to compare the ML approaches with a baseline heuristic (synonym: *Baseline model*). A baseline heuristic can be a simple ML model like a linear regression or a small Decision Tree, or alternatively, depending on the use case, it could also be a simple statement like "The next value equals the last one". The typical approach for improving ML models is to estimate the generalization error of the model on a benchmark data set when compared to a baseline heuristic. However, it is often not clear, which

baseline heuristic to consider, i.e.: The same model architecture as the benchmark model, but without tuned hyperparameters? A simple, intrinsically explainable model with or without hyperparameter tuning? A random guess? A naive guess, in which the majority class is predicted? Since we have approaches on a user-level (i.e., we consider users when splitting) and on an assessment-level (i.e., we ignore users when splitting), we also should create baseline heuristics on both levels. We additionally account for within-user variance in Ecological Momentary Assessments by averaging a user's previously known assessments. *Previously known* here means that we calculate the mode or median of all assessments of a user that are older than the given timestamp. In total, this leads to four baseline heuristics (user-level latest, user-level average, assessment-level latest, assessment-level average) that do not use any machine learning but simple heuristics. On the assessment-level, the latest known target or the mean of all known targets so far is taken to predict the next target, no matter of the user-id of this assessment. On the user-level, either the last known, or median, or mode value *of this user* is taken to predict the target. This, in turn, leads to a cold-start problem for users that appear for the first time in a dataset. In this case, either the last known, or mode, or median of all assessments that are known so far are taken to predict the target.

### ML preprocessing

Before the data and approaches could be compared, it was necessary to homogenize them. In order for all approaches to work on all data sets, at least the following information is necessary: Assessment_id, user_id, timestamp, features, and the target. Any other information such as GPS data, or additional answers to questions of the assessment, we did not include into the ML pipeline. Additionally, targets that were collected on a continuous scale, had to be binned into an ordinal scale of five classes. For an easier interpretation and readability of the outputs, we also created label encodings for each target. To ensure consistency of the pre-processing, we created helper utilities within Python to ensure that the same function was applied on each dataset. For missing values, we created a user-wise missing value treatment. More precisely, if a user skipped a question in an assessment, we filled the missing value with the mean or mode (*mode* = most common value) of all other answers of this user for this assessment. If a user had only one assessment, we filled it with the overall mean for this question.

For each dataset and for each script, we set random states and seeds to enhance reproducibility. For the outer validation set, we assigned the first 80% of all users that signed up for a study to the train set, the latest 20% to the test set. To ensure comparability, the test users were the same for all approaches. We did not shuffle the users to simulate a deployment scenario where new users join the study. This would also add potential concept drift from the train to the test set and thus improve the simulation quality.

For the cross-validation within the training set, which we call internal validation, we chose a total of 5 folds with 1 validation fold. We then applied the four baseline heuristics (on user level and assessment level with either latest target or average target as prediction) to calculate the within-train-set performance standard deviation and the mean of the weighted F1 scores for each train fold. The mean and standard deviation of the weighted F1 score are then the estimator of the performance of our model in the test set.

We call one approach superior to another if the final score is higher. The final score to evaluate an approach is calculated as:

$$f_1^{final} = f_1^{test} - \alpha \sigma(f_1^{train}) \quad (1)$$

If the standard deviation between the folds during training is large, the final score is lower. The test set must not contain any selection bias against the underlying population. The pre-factor $\alpha$ of the standard deviation is another hyperparameter. The more important model robustness for the use case, the higher $\alpha$ should be set.

### Existing train-test-split approaches

Within cross-validation, there exist several approaches on how to split up the data into folds and validate them, such as the *k*-fold approach with *k* as

## Nested 5-fold cross-validation

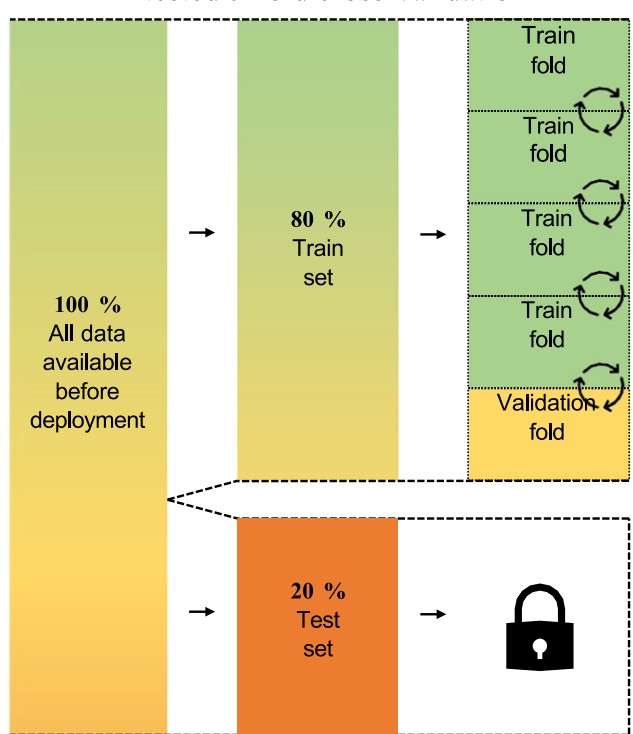

**Fig. 2 |** Schematic visualisation of the steps required to perform a *k*-fold cross-validation, here with *k* = 5.

**Stratified k-fold Cross Validation**

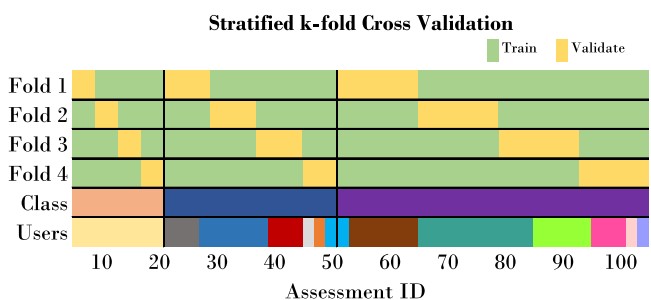

**Fig. 3 | Illustration of train-validate split for stratified 4-fold cross validation.** While this approach retains the class distribution in each fold, it still ignores user groups. Each color represents a different class or user id.

the number of folds in the training set. Here, $k - 1$ folds form the training folds and one fold is the validation fold[27]. One can then calculate *k* performance scores and their standard deviation to get an estimator for the performance of the model in the test set, which itself is an estimator for the model's performance after deployment (see also Fig. 2).

In addition, there exist the following strategies: First, (repeated) stratified *k*-fold, in which the target distribution is retained in each fold, which can also be seen in Fig. 3. After shuffling the samples, the stratified split can be repeated[3]. Second, leave-*one*-out cross-validation[28], in which the validation fold contains only *one* sample while the model has been trained on all other samples. And third, leave-*p*-out cross-validation, in which $\binom{n}{p}$ train-test-pairs are created with *n* equals number of assessments (synonym *sample*)[29].

These approaches, however, do not always focus on samples that might belong to our mHealth data peculiarities. To be more specific, they do not account for users (syn. groups, subjects) that generate daily assessments (syn. samples) with a high variance.

## Splitting approaches related to EMA

To precisely explain the splitting approaches, we would like to differentiate between the terms *folds* and *sets*. We call a chunk of samples (synonym: assessments, filled-out questionnaires) a *set* on the outer split of the data, for which we cut-off the final test *set*. However, within the training set, we then split further to create training and validation *folds*. That is, using the term *fold*, we are in the context of cross validation. When we use the term *set*, then we are in the outer split of the ML pipeline. Figure 4 visualizes this approach. Following this, we define 4 different approaches to split the data. For one of them we ignore the fact that there are users, for the other three we do not. We call these approaches *user-cut, average-user, user-wise* and *time-cut*. All approaches have in common that the first 80 % of all users are always in the training set and the remaining 20 % are in the test set. A schematic visualization of the splitting approaches is shown in Fig. 5. Within the training set, we then split on user-level for the approaches *user-cut, average-user and user-wise*, and on assessment-level for the approach *time-cut*.

In the following section, we will explain the splitting approaches in more detail. The *time-cut* approach ignores the fact of given groups in the dataset and simply creates validation folds based on the time the assessments arrive in the database. In this example, the month, in which a sample was collected, is known. More precisely, all samples from January until April are in the training set while May is in the test set. The *user-cut* approach shuffles all user *ids* and creates five data folds with distinct user-groups. It ignores the time dimension of the data, but provides user-distinct training and validation folds, which is like the GroupKFold cross-validation approach as implemented in scikit-learn[30]. The *average-user* approach is very similar to the *user-cut* approach. However, each answer of a user is replaced by the *median or mode answer* of this user up to the point in question to reduce within-user-variance. While all the above-mentioned approaches require only one single model to be trained, the *user-wise* approach requires as many models as distinct users are given in the dataset. Therefore, for each user, 80 % of his or her assessments are used to train a user-specific model, and the remaining 20% of the time-sorted assessments are used to test the model. This means that for this approach, we can directly evaluate on the test set as each model is user specific and we solved the cold-start problem by training the model on the first assessments of this user. If a user has less than 10 assessments, he or she is not evaluated on that approach.

## Ethics

Approval for the UNITI randomized controlled trial and the UNITI app was obtained by the Ethics Committee of the University Clinic of Regensburg (ethical approval No. 20-1936-101). All users read and approved the informed consent before participating in the study. The study was carried out in accordance with relevant guidelines and regulations. The procedures used in this study adhere to the tenets of the Declaration of Helsinki. The Track Your Tinnitus (TYT) study was approved by the Ethics Committee of the University Clinic of Regensburg (ethical approval No. 15-101-0204). The Corona Check (CH) study was approved by the Ethics Committee of the University of Würzburg (ethical approval no. 71/20-me) and the university's data protection officer and was carried out in accordance with the General Data Protection Regulations of the European Union. The procedures used in the Corona Health (CH) study were in accordance with the 1964 Helsinki declaration and its later amendments and was approved by the ethics committee of the University of Würzburg, Germany (No. 130/20-me). Ethical approvals include secondary use. The data from this study are available on request from the corresponding author. The data are not publicly available, as the informed consent of the participants did not provide for public publication of the data.

## Reporting summary

Further information on research design is available in the Nature Portfolio Reporting Summary linked to this article.

## Results

We will see in this results section that ignoring users in training leads to an underestimation of the generalizability of the model, the standard deviation is then too small. To further explain, a model is ranked first in the comparison of all computations if it has the highest final score, and last if it has the lowest final score. We recall the formula of the final score from the methods section: $f_1^{final} = f_1^{test} - 0.5\sigma(f_1^{train})$. For these use cases, we set

**Fig. 4 | At Start, the dataset, comprising samples with user assessments, is provided.** In the second step, users are ordered by their study registration time, with the initial 80 % designated as training users and the remaining 20 % as test users. Subsequently, assessments by training users are allocated to the training set, and those by test users to the test set. Within the training set, user grouping dictates the validation approach: group-cross-validation is applied if users are declared as a group, otherwise, standard cross-validation is utilized. We compute the average $f_1$ score, $f_1^{train}$, from training folds and the $f_1$ score on the test set, $f_1^{test}$. The standard deviation of $f_1^{train}$, $\sigma(f_1^{train})$, indicates model robustness. The hyperparameter $\alpha$ adjusts the emphasis on robustness, with higher $\alpha$ values prioritizing it. Ultimately, $f_1^{final}$, which is a more precise estimate if group-cross-validation is applied, offers a refined measure of model performance in real-world scenarios.

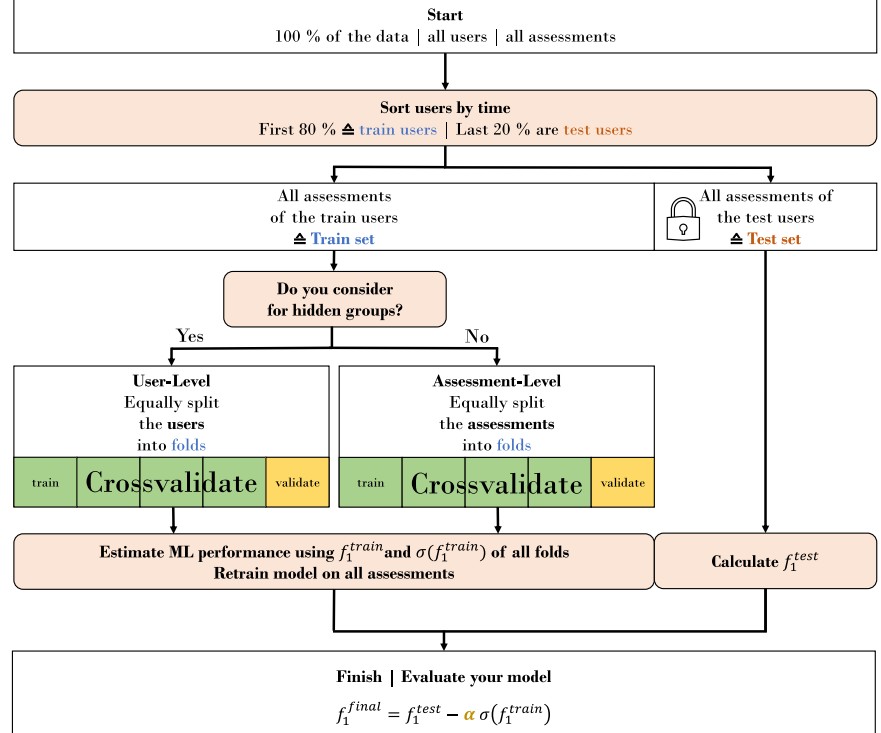

Fig. 5 | Four approaches of data splitting into train folds and validation folds within the train set. Yellow means that this sample is part of the validation fold, green means it is part of a training fold. Crossed out means that the sample has been dropped in that approach because it does not meet the requirements. Users can be sorted by time to accommodate any concept drift.

## Data splitting approaches

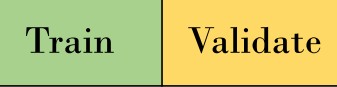

### User-cut

| Answer ID | User ID | Date | Data |
|---|---|---|---|
| 1 | U1 | Jan | ⋯ |
| 2 | U2 | Jan | ⋯ |
| 3 | U3 | Feb | ⋯ |
| 4 | U1 | Feb | ⋯ |
| 5 | U4 | Mar | ⋯ |
| 6 | U2 | Apr | ⋯ |
| 7 | U5 | Apr | ⋯ |
| 8 | U6 | May | ⋯ |
| 9 | U2 | May | ⋯ |
| 10 | U7 | May | ⋯ |

### Time-cut

| Answer ID | User ID | Date | Data |
|---|---|---|---|
| 1 | U1 | Jan | … |
| 2 | U2 | Jan | … |
| 3 | U3 | Feb | … |
| 4 | U1 | Feb | … |
| 5 | U4 | Mar | … |
| 6 | U2 | Apr | … |
| 7 | U5 | Apr | … |
| 8 | U6 | May | … |
| 9 | U2 | May | … |
| 10 | U7 | May | … |

### User-wise

| Answer ID | User ID | Date | Data |
|---|---|---|---|
| 1 | U1 | Jan | … |
| 2 | U2 | Jan | … |
| ~~3~~ | ~~U3~~ | ~~Feb~~ | ~~…~~ |
| 4 | U1 | Feb | … |
| ~~5~~ | ~~U4~~ | ~~Mar~~ | ~~…~~ |
| 6 | U2 | Mar | … |
| ~~7~~ | ~~U5~~ | ~~Apr~~ | ~~…~~ |
| ~~8~~ | ~~U6~~ | ~~May~~ | ~~…~~ |
| 9 | U2 | May | … |
| ~~10~~ | ~~U7~~ | ~~May~~ | ~~…~~ |

### Average-user

| Answer ID | User ID | Date | Data |
|---|---|---|---|
| 1 | ØU1 | Jan | … |
| 2 | ØU2 | Jan | … |
| 3 | ØU3 | Feb | … |
| 4 | ØU1 | Feb | … |
| 5 | ØU4 | Mar | … |
| 6 | ØU2 | Mar | … |
| 7 | ØU5 | Apr | … |
| 8 | ØU6 | May | … |
| 9 | ØU2 | May | … |
| 10 | ØU7 | May | … |

$\alpha = 0.5$. The greater the emphasis on model robustness and the increased concerns regarding concept drift, the greater the alpha value should be set.

### RQ1: What is the variance in performance when using different splitting methods for train and test set?

Considering performance aspects and ignoring the user groups in the data, the time cut approach has on average the best performance on assessment level. As an additional variant, we have sorted users once by time and once by random. When sorting by time, the baseline heuristic with the last known assessment of a user follows at rank 2, whereas with randomly sorted users, the user cut approach takes rank 2. The baseline heuristic with all known assessments on the user-level has the highest standard deviation in ranks, which means that this approach is highly dependent on the use case: For

some datasets, it works better, for other it does not. The *user-wise* model approach has also a higher standard deviation in the ranking score, which means that the success of this approach is more use-case specific. As we set the threshold of users to be included into this approach to a minimum of 10 assessments, we have a high chance of a selection bias for the train-test split for users with only a few assessments, which could be a reason for the larger variance in performance. Details for the result are given in Table 3.

Could there be a selection bias of users that are sorted and split by time? To answer this, we randomly draw 5 different user test sets for the whole pipeline and compared the approaches' rankings with the variation where users were sorted by time. The approaches' ranking changes by .44, which is less than one rank and can be calculated from Table 3. This shows that there is no easily classifiable group of test users.

**Table 3 | Performance comparison of cross-validated ML approaches and simple heuristics**

| | Users sorted by time | | Users split randomly | |
|---|---|---|---|---|
| | Average # | σ(#) | Average # | σ(#) |
| time_cut | 2.29 | 1.50 | 1.57 | 0.16 |
| user_cut | 3.57 | 1.72 | 3.06 | 0.11 |
| BL[4] user_based last | 3.29 | 1.70 | 3.46 | 0.21 |
| average_user | 3.86 | 0.69 | 3.51 | 0.36 |
| BL user_based all | 3.57 | 2.37 | 4.43 | 0.18 |
| user_wise | 4.33 | 2.07 | 5.10 | 0.38 |
| BL assessment_based last | 6.86 | 0.69 | 6.80 | 0.12 |
| BL assessment_based all | 7.71 | 0.49 | 7.66 | 0.15 |

Rank (#) comparison of the four splitting approaches with the four baseline heuristics, $n = 7$. Three splitting approaches are on user-level, one is on assessment level. The standard deviation is calculated from the average ranks of 7 datasets (Denoted as σ(#)). When users are not sorted by time, the approaches are more robust in their rankings, which means that the user cut approach is more likely to work consistently better than the baseline heuristic on user-level.[4]

[4]BL = Baseline

Cross-validation within the train helps to estimate the generalization error of the model for unseen data. On assessment-level, the standard deviations of the weighted F1 score within the train set for all datasets varies between 0.25 % for TrackYourTinnitus and 1.29 % for Corona Health Stress. On user-level, depending on the splitting approach, the standard deviation varies from 1.42 % to 4.69 %. However, on the test set, the estimator of the generalization error (i.e., the standard deviation of the F1 scores of the validation folds within the train set) is too low for all 7 datasets on assessment-level. On user-level, the estimator of the generalization error is too low for 4 out of 7 datasets. We define the estimator of the generalization error as *in range* if its smaller or equals the performance drop between validation and test set. Details for the result are given in Table 4.

Both approaches, user- and assessment, overestimate the performance of the model during training. However, the quality of estimator of the generalization error increases if users are split on user-level.

### RQ2: In which cases is the development, deployment and maintenance of a ML model compared to a simple baseline heuristic worthwhile?

For our 7 datasets, the baseline heuristics on a user-level perform better than those on assessment-level. For the datasets *Corona Check (CC), Corona Health Stress (CH), TrackYourTinnitus (TYT)* and *UNITI*, the last known user assessment is the best predictor within the baseline heuristics. For the psychological Corona Health study with adolescents (CHY) and adults (CHA), and physical health for adults (CHP), the average of the historic assessments is the best baseline predictor. The last known assessment on an assessment-level as a baseline heuristic performs worse for each dataset compared to the assessment level. The average of all so far known assessment as a predictor for the next assessment - independent from the user - has worst performance within the baseline heuristics for all datasets except CHA. Notably, the larger the number of assessments, the more the all-instances-approach on assessment-level converts to the mean of the target, which has high bias and minimum variance.

These results lead us to conclude that recognizing user groups in datasets leads to an improved baseline when trying to predict future ones from historical assessments. When these non-machine-learning baseline heuristics are then compared to machine learning models without hyper-parameter tuning, it is found that they sometimes outperform or similarly outperform the machine learning model.

The approaches ranking in Table 5 shows the general overestimation of the performance of the *time-cut* approach as this approach is ranked best on average. It can be also seen that these approaches are ranked closely to each other. We chose $\alpha$ to be 0.5. Because we only subtract 0.5 (0.5 = $\alpha$, our

**Table 4 | Performance scores by study**

| Study | Score | User-Level | | Assessment-Level |
|---|---|---|---|---|
| | | user_cut | average_user | time_cut |
| CC | Std Validation | 1.42% | 4.69% | 0.54% |
| | F1 Validation | 76.80% | 72.50% | 77.57% |
| | F1 Test | 67.54% | 64.60% | 67.98% |
| | Performance Drop | −9.27% | −7.90% | −9.59% |
| CHS | Std Validation | 4.95% | 3.59% | 1.29% |
| | F1 Validation | 54.73% | 51.19% | 57.47% |
| | F1 Test | 51.32% | 53.10% | 53.15% |
| | Performance Drop | −3.41% | 1.91% | −4.31% |
| CHY | Std Validation | 1.80% | 1.46% | 0.71% |
| | F1 Validation | 98.85% | 98.28% | 98.89% |
| | F1 Test | 97.63% | 94.86% | 98.18% |
| | Performance Drop | −1.21% | −3.42% | −0.71% |
| CHA | Std Validation | 3.75% | 3.79% | 1.23% |
| | F1 Validation | 65.80% | 66.73% | 69.87% |
| | F1 Test | 61.79% | 62.24% | 63.05% |
| | Performance Drop | −4.00% | −4.48% | −6.81% |
| CHP | Std Validation | 2.24% | 2.47% | 1.06% |
| | F1 Validation | 47.79% | 43.70% | 53.25% |
| | F1 Test | 45.38% | 46.53% | 45.97% |
| | Performance Drop | −2.40% | 2.84% | −7.28% |
| TYT | Std Validation | 2.25% | 3.78% | 0.25% |
| | F1 Validation | 54.88% | 45.97% | 58.70% |
| | F1 Test | 56.26% | 40.57% | 57.26% |
| | Performance Drop | 1.38% | −5.40% | −1.44% |
| UNITI | Std Validation | 2.51% | 1.62% | 0.36% |
| | F1 Validation | 55.81% | 55.92% | 58.30% |
| | F1 Test | 52.00% | 50.12% | 53.07% |
| | Performance Drop | −3.82% | −5.80% | −5.23% |

Performance scores and standard deviations of the seven use cases on user- and assessment-level. As we used a 5-fold cv approach, $n = 5$. For the user-level, there are two splitting approaches shown: *User-cut*, with users sorted by time of sign-up, and *average-user*, where an answer given by a specific user is averaged with the users' previously given answers. $f_1^{train}$ conforms the average $f_1$ scores of the validation folds of the train set.[6]

[6]Studies are abbreviated as follows: Unification of Treatments and Interventions for Tinnitus Patients (UNITI), Track Your Tinnitus (TYT), Corona Health Physical Health Adults (CHP), Corona Health Mental Health Adults (CHA), Corona Health Mental Health Adolescents (CHY), Corona Health Stress (CHS), and Corona Check (CC).

hyperparameter to control importance of model robustness) of the standard deviation of the $f_1$ scores of the validation folds, approaches with a higher standard deviation are less punished. This means, in turn, that the over-estimation of the performance of the splits on assessment-level would be higher if $\alpha$ was higher. Another reason for the similarity of the approaches is that the same model architecture has been finally trained on all assessments of all train users to be evaluated on the test set. Thus, the only difference of the rankings results from the standard deviation of the $f_1$ scores of the validation folds.

To answer the question whether it is worthwhile to turn a prediction task into an ML project, further constraints should be considered. The above

**Table 5 | Average Ranking Per Approach**

| Kind of model | ML | BL | BL | ML | ML | ML | BL | BL |
|---|---|---|---|---|---|---|---|---|
| Approach | Time Cut | User-Level | User-Level | User Cut | Avg. User | User Wise | A.-Level | A.-Level |
| | | Last | All inst. | | | | | |
| Average# | 2.29 | 3.29 | 3.57 | 3.57 | 3.86 | 4.33 | 6.86 | 7.71 |
| σ(#) | 1.5 | 1.7 | 2.37 | 1.72 | 0.69 | 2.07 | 0.69 | 0.49 |

Average rank (*n* = 7) of the approach for all datasets, including the standard deviation of the rank one line below. On average, the baseline heuristics on the user-level are ranked slightly better than the ML model on a user-level. Best rank is left, worst rank is right. Headers in bold.[8]

[8]A. = Assessment, # = Rank, σ(#) = Standard deviation of #., Avg. = Average., inst. = Instances.

analysis shows that the baseline heuristics are competitive to the non-tuned random forest with much lower complexity. At the same time, the overall results are an f1 score between 55 and 65 for a multi-class classification with potential for improvement. Thus, the question should be additionally asked, from which $f_1$ score can be deployed, which depends on the use case, and in addition it is not clear whether the ML approach can be significantly improved by a different model or the right tuning.

## Discussion

The present work compared the performance of a tree-based ensemble method if the split of the data happens on two different levels: User and assessment. It further compared this performance to non-ML approaches that uses simple heuristics to also predict the target on a user- or assessment level. We quickly summarize the findings and then discuss them in more detail in the sections below. Neglecting user data during cross-validation may result in an inflated estimation of model performance and robustness, a phenomenon critical to the integrity of model evaluation. In specific scenarios, empirical evidence suggests that straightforward heuristic approaches can rival the efficacy of complex tree-based ensemble methodologies. Particularly, heuristics tailored or applied at the user level manifest a distinct advantage, while machine learning models maintain efficacy at the assessment level. Additionally, the methodological sorting of users in the dataset can serve as a proxy for concept drift in longitudinal studies, given a sufficiently extensive data collection period. This manipulation affects the test set outcomes, underscoring the influence of temporal user behavior variations on model validation.

The - still - small number of 7 use cases itself has a risk of selection bias in the data, features, or variables. This limits the generalizability of the statements. However, it is also arguable whether the trends found turn in a different direction when more use cases are included in the analysis. We do not believe that the tendencies would turn. We restricted the ML model to be a random forest classifier with a default hyperparameter set up to increase the degree of comparability between use cases. We are aware that each use case is different and direct comparability is not possible. Furthermore, we could have additionally evaluated the entire pipeline on other ML models that are not tree-based. However, this would have added another dimension to the comparison and further complicated the comparison of the results. Therefore, we cannot preclude that the results would have been substantially different for non-tree-based methods, which can be investigated further in future analyses.

Future research of this user-vs.-assessment-level comparison could include a hyperparameter tuning of the model on each use case, a change of model kind (i.e., from a random forest to a support vector machine) to see whether this changes the ranking. The overarching goal remains to obtain the most accurate estimate of the model's performance after deployment.

We cannot give a final answer to what can be chosen as a common baseline heuristic. In machine learning projects, a majority vote is typically used for classification tasks, and a simple model such as a linear regression can be used for regression tasks. These approaches can also be called naive approaches since they often do not do justice to the complexity of the use case. Nevertheless, the power of a simple non-ML heuristic should not be underestimated. If only a few percentage points more performance can be achieved by the maintenance- and development-intensive ML approach, it is worth considering whether the application of a simple heuristic such as "the next assessment will be the same as the last one" is sufficient for a use case. Notably, Cawley and Talbot argue that it might be easier to build domain expert knowledge into hierarchical models, which could also function as a baseline heuristic[10].

To retain consistency and reproducibility, we kept the users sorted by sign-up date to draw train and test users. The advantage of sorting the users is that one can simulate potential concept drift during training. The disadvantage, however, is an inherent risk of a selection bias towards users that signed up earlier for a study. From Figure 3, we can see that the overfitting of users increases when we shuffle them. We conclude this from the fact that the difference between the average ranks of the approaches *time cut* and *user cut* increases. The advantage of shuffling users is that the splitting methods seem to depend less on the dataset. This can be deduced from the reduced standard deviation of the ranks compared to the sorted users.

Regardless of the level of splitting (user- or assessment-level), one can expect a performance drop if unknown users with unknown assessments are withheld from the model in the test set. When splitting at the user-level, the performance drop is lower during training and validation compared to the assessment-level. However, it remains questionable why we see this performance drop in the test set at all, because both, the validation folds and the test set contain unknown users with unknown assessments. A possible cause could be simple overfitting of the training data with the large random forest classifier and its 100 trees. But, also a single tree with max depth = number of features and balanced class weights has this performance drop from the validation to the test set. One explanation for the defiant performance drop could be that during cross validation information leaks from training folds to validation folds, but not to the test set.

## Conclusions

A simple heuristic is not always trivial to beat by an ML model, depending on the use case and the complexity of the search space. Thinking of the complexity that a ML model adds to a project, a heuristic might be a valuable start to see how well the model fits into the workflow and improves the outcome. A frequent communication with the domain expert of the use case helps to set up a heuristic as a baseline heuristic. In a second step, it can be evaluated whether the performance gain from an ML model justifies the additional development effort.

## Data availability

In relation to the individual data sets used (see Table 2), the availability is as follows: (1) TYT: The data presented in this study are available on request from the corresponding author. The data are not publicly available for data protection reasons. (2) UNITI, Corona Check, Corona Health: The investigators have access to the study data. Raw data (de-identified) can be made available on request from the corresponding author. Furthermore, only the mHealth data was used in this study on UNITI, but the entire UNITI RCT study contains even more data, which can be found here[20].

## Code availability

All code to replicate the results, models, numbers, figures, and tables is publicly available to anyone on https://github.com/joa24jm/UsAs[32], DOI = 10.5281/zenodo.10401660.

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

## Acknowledgements

This work was partly funded by the ESIT (European School for Interdisciplinary Tinnitus Research[31]) project, which is financed by European Union's Horizon 2020 research and innovation programme under the Marie Sklodowska-Curie grant agreement number 722046 and the UNITI (Unification of Treatments and Interventions for Tinnitus Patients) project financed by the European Union's Horizon 2020 Research and Innovation Programme, Grant Agreement Number 848261[19]. J.A. and R.P. are supported by grants in the projects COMPASS and NAPKON. The COMPASS and NAPKON projects are part of the German COVID-19 Research Network of University Medicine ("Netzwerk Universitätsmedizin"), funded by the German Federal Ministry of Education and Research (funding reference 01KX2021). This publication was supported by the Open Access Publication Fund of the University of Wuerzburg.

## Author contributions

J.A. primarily wrote this paper, created the figures, tables and plots, and trained the machine learning algorithms. R.P. supervised and revised the paper.

## Funding

## Competing interests

The authors declare no competing interests.
