## [Peer Review File · Communications Medicine]

Reviewers' comments:

Reviewer #1 (Remarks to the Author):

The article titled "7 observational mHealth studies and 10 years of experience: Can ignoring groups in Machine Learning pipelines lead to overestimation of model performance? Analyses of group-wise validation as well as baseline and concept-drift considerations" discusses the importance of choosing an appropriate test set for evaluating the performance of machine learning models, particularly in the mHealth domain where data is often collected through Ecological Momentary Assessment (EMA). The study compares different train-test-split methods applied at group and assessment levels and evaluates the variance of model performances using four train-test split approaches and four baseline heuristics. The results suggest that the assessment level split can lead to an overestimation of ML performance after deployment, and completion behavior within a user varies little for individual questions. The article concludes that more analysis is needed to obtain robust ML models in the mHealth domain, and domain experts should be consulted to take hidden groups into account when splitting the data. Simple heuristics can also be considered as alternatives to complex ML models. Overall, the paper is written with a high degree of proficiency, and it conveys a wealth of information. There are a few minor concerns that the authors may wish to address.

- Minor Concerns

1. The abstract and its elements require modification, as the background, methods, results, and conclusion do not effectively demonstrate the content of the paper.
2. The introduction section of the paper could benefit from more detailed and informative content that adequately prepares the reader for the main contributions of the study. There appear to be some gaps between the related works, which could be made more evident with greater attention to detail. It is recommended that the authors strive to enhance the consistency of the introduction by elaborating on the goals and objectives of the paper.
3. In Section 1.1, the sentence "In addition, there exists stratified k-fold, repeated stratified k-fold, leave-one-out cross-validation [7], and leave-p-out cross-validation, ..." need to be modified because some methods are listed and the very last one leave-p-out cross-validation is described very briefly. For the sake of consistency, authors may add brief description to other stratified k-fold, repeated stratified k-fold, leave-one-out cross validation.
4. On page 4, there is a reference missing for the sentence 'Refaeilzadeh et al. provide an overview.' The authors should add a reference for this statement.
5. In "The features are a list of Boolean variables, which where known at this time to be typically related with a Covid infection, such as fever", there is an error in the use of "where" in the first sentence. It should be replaced with "were," making the sentence grammatically correct as: "The features are a list of Boolean variables, which were known at this time to be typically related with a Covid infection, such as fever, a sore throat, a runny nose, cough, loss of smell, loss of taste, shortness of breath, headache, muscle pain, diarrhea, and general weakness."
6. In Section 4, the sentence "Ignoring users in datasets during cross-validation leads to an overestimation of the model's performance and the model's robustness" can be replaced with "Ignoring users in datasets during cross-validation leads to an overestimation of the model's performance and robustness."
7. In Section 4, in the sentence "For some use cases, simple heuristics are as good as complicated tree based ensemble methods. Within this domain, heuristics are more auspicious if they are trained or applied at the user level. ML models also work at the assessment level" the authors may replace "auspicious" with "advantageous." Auspicious" is a valid word, meaning favorable or promising. However, in the given context, it seems like the intended word was "advantageous", which means beneficial or helpful.
8. "Is" should be changed to "it" in the sentence "Notably, Cawley and Talbot argue that is might be easier to build domain expert knowledge into hierarchical models, which could also function as a baseline heuristic." The corrected sentence would read: "Notably, Cawley and Talbot argue that it

might be easier to build domain expert knowledge into hierarchical models, which could also function as a baseline heuristic."

Reviewer #2 (Remarks to the Author):

1. Overall work is good.
2. Carefully review the manuscript for any instances of missing or incorrect words. In Section 2.5, there is an error where the line begins with "For the outer validation set, we...set set." It should be "test set" instead of "set." Please ensure that this correction is made accordingly.
3. Ensure the manuscript is free from grammatical issues.
4. It would be great if it includes a figure that provides an outline of the overall procedures.
5. I have a little confusion on the figure 5 writing. It said "All assessments of 80 % of the train users". You choose 80% of total data as a train and the rest 20% as a test. Please double check the meaning of these two lines and correct them accordingly if needed.
6. Did you use any feature selection techniques for any dataset?
7. Determine the appropriate number of epochs or iterations and randomly select train and test data from the dataset. Finally, present the average results derived from these iterations.
8. In the section discussing the limitations of this study, it is important to include any potential influencing factors and outline plans for future improvements in the latter part of the Discussion section.

Reviewer #3 (Remarks to the Author):

When applying machine learning algorithms to medical data, subject-specific characteristics may add difficulty or cause trouble if the test set is not selected appropriately. In mHealth-related studies, large numbers of assessment occur on the same user, while the numbers are different across users. As a result, user heterogeneity is very important, and should be taken into consideration when splitting the training and testing datasets. In this paper, the authors studied various train-test-split approaches, and paid special attention to time effect, which fills the gap of the area. The authors also compared the machine learning methods to some simple baseline heuristic, and find that the heuristic approaches can beat the ML approaches in certain circumstances. The authors, consequently, suggested to pay more attention to heuristic approaches. This research fills the gap in mHealth with the study on train-test-split approaches. The authors find promising and valuable results by evaluating different approaches. Results will contribute to the field greatly. However, the paper is a little difficult to read, especially to readers outside the specific area. Some simple explanations on the background of the studies will help a lot instead of asking the readers to read the references. Here are some detailed comments:

1. In the machine learning preprocessing, the authors estimate missing values with mean or mode of other assessments from the same user. Is it possible to just keep the data missing and consider the particular assessment having longer time interval?
2. For the outer validation set, only splitting on users are evaluated. How would the results change or

not change if splitting on assessments?

3. What is the rationale behind the evaluation function:

$$f_{\text{final}} = f_{\text{test}} -$$

$0.5\sigma(f_{\text{train}})$? Why does it make sense to choose an arbitrary number 0.5?

4. When introducing the background information for ML, it may be more clear if the authors introduce the name of the 7 studies before directly pointing them with descriptions as "This can happen several times a day (e.g., TYT) or at weekly intervals (e.g., studies in the Corona Health app)." There are several other places with similar issue.

5. There are some typos. For example, on Page 3 last paragraph, in sentence "One can than calculate k performance scores and their standard deviation to...", "than" should be changed to "then".

Point-by-Point-response

Reviewer #1 (Remarks to the Author):

1. The abstract and its elements require modification, as the background, methods, results, and conclusion do not effectively demonstrate the content of the paper.

We updated the abstract. Now it hopefully reflects the content of our paper better.

Background. Machine learning (ML) models are evaluated in a test set to estimate model performance after deployment. The design of the test set is therefore of importance because if the data distribution after deployment differs too much, the model performance decreases. At the same time, the data often contains undetected groups. For example, multiple assessments from one user may constitute a group, which is usually the case in mHealth scenarios.

Methods. In this work, we evaluate a model's performance using several cross-validation train-test-split approaches, in some cases deliberately ignoring the groups. By sorting the groups (in our case: users) by time, we additionally simulate a concept drift scenario for better external validity. For this evaluation, we use 7 longitudinal mHealth datasets, all containing Ecological Momentary Assessments (EMA). Further, we compared the model performance with baseline heuristics, questioning the essential utility of a complex ML model.

Results. Hidden groups in the dataset leads to overestimation of ML performance after deployment. For prediction, a user's last completed questionnaire is a reasonable heuristic for the next response, and potentially outperforms a complex ML model. Because we included 7 studies, low variance appears to be a more fundamental phenomenon of mHealth datasets.

Conclusion. The way mHealth-based data are generated by EMA leads to questions of user and assessment level and appropriate validation of ML models. Our analysis shows that further research needs to follow to obtain robust ML models. In addition, simple heuristics can be considered as an alternative for ML. Domain experts should be consulted to find potentially hidden groups in the data.

2. The introduction section of the paper could benefit from more detailed and informative content that adequately prepares the reader for the main contributions of the study. There appear to be some gaps between the related works, which could be made more evident with greater attention to detail. It is recommended that the authors strive to enhance the consistency of the introduction by elaborating on the goals and objectives of the paper.

Thank you for this hint! We used these four papers to close the gap between our main contribution and the related work:

and the appreciation of machine learning based classification and raise new questions about this. To this end, we will briefly explain the background. Using simpler, interpretable models instead of complex ML models is Using simple, understandable models rather than complex black box models is a clamor of Rudin et. al. which motivates us to evaluate simple heuristics against complex models-\cite{rudin2019stop}.

motivates us to evaluate simple heuristics against complex models-\cite{rudin2019stop}. The Cross-Industry Standard Process for Data Mining (CRISP-DM) highlights the importance of subject matter experts to get familiar with a dataset-\cite{chapman2000crisp}. In turn, familiarity with the dataset

different variants of cross validation. It is well shown that the type of validation can cause extreme differences. In older work, different cross validation techniques were evaluated on datasets with different recommendations on the number of optimal folds~\cite{kohavi1995study, dietterich1998approximate}. We

- [1] Rudin, C. (2019). *Stop explaining black box machine learning models for high stakes decisions and use interpretable models instead*. *Nature machine intelligence*, 1(5), 206-215.
- [2] Chapman, P., Clinton, J., Kerber, R., Khabaza, T., Reinartz, T., Shearer, C., & Wirth, R. (2000). *CRISP-DM 1.0: Step-by-step data mining guide*. SPSS inc, 9(13), 1-73.
- [3] Kohavi, R. (1995, August). *A study of cross-validation and bootstrap for accuracy estimation and model selection*. In *Ijcai (Vol. 14, No. 2, pp. 1137-1145)*.
- [4] Dietterich, T. G. (1998). *Approximate statistical tests for comparing supervised classification learning algorithms*. *Neural computation*, 10(7), 1895-1923.

3. In Section 1.1, the sentence “In addition, there exists stratified k-fold, repeated stratified k-fold, leave-one-out cross-validation [7], and leave-p-out cross-validation, ...” need to be modified because some methods are listed and the very last one leave-p-out cross-validation is described very briefly. For the sake of consistency, authors may add brief description to other stratified k-fold, repeated stratified k-fold, leave-one-out cross validation.

Thank you for this hint. We added brief descriptions and a schematic visualization for an easier understanding.

- (Repeated) stratified k -fold, where the target distribution is retained in each fold. After shuffling the samples, the stratified split can be repeated [3].
- Leave-*one*-out cross-validation [11], where the validation fold contains only *one* sample while the model has been trained on all other samples.
- Leave- p -out cross-validation, where $\binom{n}{p}$ train-test-pairs are created with n equals number of assessments (synonym *sample*) [12].

Figure 1: Illustration of train-validate split for stratified 4-fold cross validation. While this approach retains the class distribution in each fold, it still ignores user groups. Each color represents a different class or user id.

4. On page 4, there is a reference missing for the sentence 'Refaeilzadeh et al. provide an overview.' The authors should add a reference for this statement.

The citation for this reference is in the next sentence as we briefly explain the contribution of this paper. We added a second citation:

Refaeilzadeh et. al. give an overview of common cross-validation techniques such as leave-one-out, repeated k-fold, or hold-out validation~\cite{refaeilzadeh2009cross}. They discuss pros and cons of each kind and also mention an \textit{underestimated performance variance} for repeated k-fold cross-validation, but they also do not address the problem with (unknown) groups in the dataset~\cite{refaeilzadeh2009cross}.

5. In “The features are a list of Boolean variables, which where known at this time to be typically related with a Covid infection, such as fever”, there is an error in the use of "where" in the first sentence. It should be replaced with "were," making the sentence grammatically correct as: "The features are a list of Boolean variables, which were known at this time to be typically related with a Covid infection, such as fever, a sore throat, a runny nose, cough, loss of smell, loss of taste, shortness of breath, headache, muscle pain, diarrhea, and general weakness."

Thank you for spotting this typo. We corrected it.

6. In Section 4, the sentence “Ignoring users in datasets during cross-validation leads to an overestimation of the model’s performance and the model’s robustness” can be replaced with “Ignoring users in datasets during cross-validation leads to an overestimation of the model’s performance and robustness.”

Thank you for shortening this sentence. We changed it.

7. In Section 4, in the sentence “For some use cases, simple heuristics are as good as complicated tree based ensemble methods. Within this domain, heuristics are more auspicious if they are trained or applied at the user level. ML models also work at the assessment level” the authors may replace “auspicious” with "advantageous." Auspicious" is a valid word, meaning favorable or promising. However, in the given context, it seems like the intended word was "advantageous", which means beneficial or helpful.

Thank you for the clarification. We changed the wording accordingly.

8. "Is" should be changed to "it" in the sentence "Notably, Cawley and Talbot argue that is might be easier to build domain expert knowledge into hierarchical models, which could also function as a baseline heuristic." The corrected sentence would read: "Notably, Cawley and Talbot argue that it might be easier to build domain expert knowledge into hierarchical models, which could also function as a baseline heuristic."

Thank you for spotting this typo. We corrected it.

Reviewer #2 (Remarks to the Author):

1. Overall work is good.

We thank you very much for this compliment.

2. Carefully review the manuscript for any instances of missing or incorrect words. In Section 2.5, there is an error where the line begins with "For the outer validation set, we...set set." It should be "test set" instead of "set." Please ensure that this correction is made accordingly.

Thank you for this hint! We corrected the specific sentence and reviewed the whole script.

3. Ensure the manuscript is free from grammatical issues.

We asked an external third to additionally review the manuscript for typos or grammatical issues.

4. It would be great if it includes a figure that provides an outline of the overall procedures.

Thank you for this feedback. It seems that Figure 5 (which is now Figure 4 as we updated the manuscript) does not provide enough detail yet. We thus extended figure 6:

5. I have a little confusion on the figure 5 writing. It said “All assessments of 80 % of the train users”. You choose 80% of total data as a train and the rest 20% as a test. Please double check the meaning of these two lines and correct them accordingly if needed.

Thank you for paying attention to this sentence. It is probably one of the most important sentences in the manuscript and it is indeed correct: At the beginning, all data of all users is given. We then split 80 % of the users (not assessments) to create a train set. As each user fills out a different number of assessments, this train set does not necessarily contain 80 % of the data.

However, we agree with you that this sentence is still confusing, which is why we updated the wording: “All assessments of the train users” form the train set, and 80 % of all users are the train users.

6. Did you use any feature selection techniques for any dataset?

That is a good hint. However, we have consciously not done this, because the focus of this work is on data splitting and baseline heuristics, not feature selection. We selected the features according to CRISP-DM [1] in consultation with the respective subject matter experts of the studies by asking them about good features for the respective target. We substantiated the feature selection with relevant literature.

[1] Wirth, Rüdiger, and Jochen Hipp. "CRISP-DM: Towards a standard process model for data mining." *Proceedings of the 4th international conference on the practical applications of knowledge discovery and data mining*. Vol. 1. 2000.

7. Determine the appropriate number of epochs or iterations and randomly select train and test data from the dataset. Finally, present the average results derived from these iterations.

We would kindly ask you to look at our updated Figure 6: The dataset has two levels, users, and assessments. To ensure comparability, all approaches must be evaluated on the same set of users. Additionally, and to simulate concept drift, we split them along the time-axis: The first 80 % of the users that sign up are assigned to the train set, the last 20 % to the test set. No shuffling or random selection at this stage of the pipeline.

You are now rightly asking whether this could have created a simple test set (= a biased test set) by chance. And we had addressed this in the Methods section with a short comment that we cross checked this by creating another 5 different user test sets which were then passed through the whole pipeline. The rankings of the approaches changed by .44 on average which shows us that the user test set does not happen to be easy to classify by chance, no selection bias here! We extended the sentence in the manuscript describing this:

Could there be a selection bias of users that are sorted and split by time? To answer this, we randomly draw 5 different user test sets for the whole pipeline and compared the approaches' rankings with the variation where users were sorted by time. The approaches' ranking changes by .44, which is less than one rank and can be calculated from Table 1. This shows that there happens to be no easily classifiable group of test users.

We provide further details in Table 3:

	Users sorted by time		Users split randomly	
	Average rank	Std of average rank	Average rank	Std of average rank
time_cut	2,29	1,50	1,57	0,16
user_cut	3,57	1,72	3,06	0,11
BL user_based last	3,29	1,70	3,46	0,21
average_user	3,86	0,69	3,51	0,36
BL user_based all	3,57	2,37	4,43	0,18
user_wise	4,33	2,07	5,10	0,38
BL assessment_based last	6,86	0,69	6,80	0,12
BL assessment_based all	7,71	0,49	7,66	0,15

Table 3: Rank comparison of the four splitting approaches with the four baseline heuristics. Greener means better. Three splitting approaches are on user-level, one is on assessment level. The standard deviation is calculated from the average ranks of 7 datasets. When users are not sorted by time, the approaches are more robust in their rankings, which means that the user cut approach is more likely to work consistently better than the baseline heuristic on user-level. BL = Baseline.

We also kindly point to the discussion section:

`\subsection{The impact of shuffled users}` In order to retain consistency and reproducibility, we kept the users sorted by sign-up date to draw train and test users. The advantage of sorting the users is that one can simulate potential concept drift during training. The disadvantage, however, is an inherent risk of a selection bias towards users that signed up earlier for a study. From Table `\ref{tab:final_result}`, we can see that the overfitting of users increases when we shuffle them. We conclude this from the fact that the difference between the average ranks of the approaches `\textit{time cut}` and `\textit{user cut}` increases. The advantage of shuffling users is that the splitting methods seem to depend less on the dataset. This can be deduced from the reduced standard deviation of the ranks compared to the sorted users.

8. In the section discussing the limitations of this study, it is important to include any potential influencing factors and outline plans for future improvements in the latter part of the Discussion section.

In the "Limitations" subsection, we have discussed any influencing factors that we believe limit our results. We have not been able to think of any other limitations, but we are open to further suggestions for limitations.

For added a paragraph with Future Research recommendations.

`\paragraph{Future research}` of this user-vs.-assessment-level comparison could include a hyperparameter tuning of the model on each use case, a change of model kind (i.e., from a random forest to a support vector machine) to see whether this changes the ranking. The overarching goal remains to obtain the most accurate estimate of the model's performance after deployment.

To date, we have no ideas for concrete further improvements but the strong recommendation to consult domain experts to find hidden groups and first consider baseline heuristics before diving into the complex process of training an ML model.

Reviewer #3 (Remarks to the Author):

When applying machine learning algorithms to medical data, subject-specific characteristics may add difficulty or cause trouble if the test set is not selected appropriately. In mHealth-related studies, large numbers of assessment occur on the same user, while the numbers are different across users. As a result, user heterogeneity is very important, and should be taken into consideration when splitting the training and testing datasets. In this paper, the authors studied

various train-test-split approaches, and paid special attention to time effect, which fills the gap of the area. The authors also compared the machine learning methods to some simple baseline heuristic, and find that the heuristic approaches can beat the ML approaches in certain circumstances. The authors, consequently, suggested to pay more attention to heuristic approaches. This research fills the gap in mHealth with the study on train-test-split approaches. The authors find promising and valuable results by evaluating different approaches. Results will contribute to the field greatly.

However, the paper is a little difficult to read, especially to readers outside the specific area. Some simple explanations on the background of the studies will help a lot instead of asking the readers to read the references. Here are some detailed comments:

We provided some more details about the included mHealth studies, so the interested reader does not have to look up the original study papers. We also provided an overview table:

Mobile Application	 Track Your Tinnitus TYT	 Corona Check CC	 Corona Health CH	 Unification of Treatments and Interventions for Tinnitus Patients UNITI
Studies Involved & Background	TYT was launched in 2014 to find out more about daily tinnitus fluctuations and since then has been a longitudinal observational study that we initiated without a project background using an iOS and Android app in the official app stores.	During the Covid 19 pandemic, Covid testing, and knowledge of the virus were scarce. Thus, we developed the CC app to provide feedback based on their reported symptoms.	 • Mental health for  • adults (CHA) • adolescents (CHY) • Physical health for  • adults (CHP) • Stress (CHS) This app contains mental and physical health studies where user behavior can be tracked during the Covid pandemic.	UNITI wants to develop a model that enables patient-specific treatment of tinnitus.
Project Partners	Tinnitus Research Initiative	Bavarian State Office for Health and Food Safety	Robert Koch Institute	European Union's Horizon 2020 Research and Innovation Programme

Table 1: Overview of the mobile applications and the studies involved in this project: TrackYourTinnitus [21], Corona Check [23], Corona Health [6], and Unification of Treatments and Interventions for Tinnitus Patients [22]

`\paragraph{TrackYourTinnitus (TYT)}`

With this app it is possible to record the individual fluctuations in tinnitus perception. With the help of a mobile device, users can systematically measure the fluctuations of their tinnitus. Via the `\href{www.trackyourtinnitus.com}{TYT website}`, users can then view the progress of their own data and, if necessary, discuss it with their physician.

`\paragraph{Unification of Treatments and Interventions for Tinnitus Patients (UNITI)}`

The overall goal of UNITI is to treat the heterogeneity of tinnitus patients on an individual basis. This requires understanding more about the patient-specific symptoms that are captured by EMA in real time.

`\paragraph{Corona Check (CC)}`

At the beginning of the COVID-19 pandemic, it was not easy to get initial feedback about an infection, given the lack of knowledge about the novel virus and the absence of widely available tests. To assist all citizens in this regard, we launched the mobile health app Corona Check together with the Robert-Koch-Institute from Germany~\cite{beierle2023self}.

`\paragraph{Corona Health | Mental health for adults (CHA)}`

Within the Corona Health app, users can enroll in different studies that interest them and to which they would like to contribute. In `\textit{Mental health for adults}`, this is adult mental health, which is evaluated with standard instruments.

`\paragraph{Corona Health | Mental health for adolescents (CHY)}`

Similar to the adult cohort, the mental health of adolescents during the pandemic and its lock-downs is also captured by our app using EMA.\\

A lightweight version of the mental health questionnaire for adults was also offered to

`\paragraph{Corona Health | Physical health for adults (CHP)}`

Analogous to the mental health of adults, this study aims to track how the physical health of adults changes during the pandemic period.\\

1. In the machine learning preprocessing, the authors estimate missing values with mean or mode of other assessments from the same user. Is it possible to just keep the data missing and consider the particular assessment having longer time interval?

This is an interesting idea. However, this is technically not possible. The ML model requires all numbers to be there, N/A is not calculable. So, the only alternative would be to just wait until the user hopefully answers the question the next time she fills out the assessment. But this would mean to skip one prediction and that is one datapoint less to evaluate.

The other alternative is to wait for the next data points to feed this into the model instead of the last data points, but this would make the prediction task obsolete: For example, if we want to predict tinnitus occurrence on April 15th with data from April 7th, but the data from April 7th is missing, we cannot wait until April 15th for the next data points because then we would not predict anything anymore.

2. For the outer validation set, only splitting on users are evaluated. How would the results change or not change if splitting on assessments?

If we split on assessment level on the outer test set (we use the term validation within the train set when performing cross validation), we would allow one user to be represented in both, train, and test sets. But that's the key point of the whole paper: In a real-world scenario, a ML model is trained on users that entered the study in a specific period, let's say, January to August. If we deploy in August, we use the model on users that appear in September onwards. So, during training, it's not possible to train the model with a user from September.

However, within the train set, we can split on assessment level and evaluate this against splitting on user level because – having the real-world scenario in mind – we have all users from January to August and we can allow users to be represented in different folds (which we do not recommend).

The results when doing that are known to us:

The model overfits on strongly represented users that have assessments in all folds during cross-validation. So, the changed results are:

- Overfitting on users
- Overestimation of deployment performance
- Underestimation of performance variance

3. What is the rationale behind the evaluation function: $f_1^{\text{final}} = f_1^{\text{test}} - 0.5\sigma(f_1^{\text{train}})$? Why does it make sense to choose an arbitrary number 0.5?

The 0.5 depends on the use case. First, we state that one model is superior to another if it has a higher f_1^{final} score. We then penalize the model if the variance of the performance between the train folds is large. The more important a low variance in model performance is to us, the larger we make the 0.5 and vice versa. We discussed this in the methods section:

We call one approach superior to another if the final score is higher. The final score to evaluate an approach is calculated as:

$$f_1^{\text{final}} = f_1^{\text{test}} - 0.5 \sigma(f_1^{\text{train}})$$

If the standard deviation between the folds during training is large, the final score is lower. The test set must not contain any selection bias against the underlying population. The prefactor of the standard deviation σ with 0.5 has been chosen arbitrarily. It should be set higher the more important the generalization error of the model is; i.e., models with high performance variance between validation folds during training will receive an even lower final score.

4. When introducing the background information for ML, it may be more clear if the authors introduce the name of the 7 studies before directly pointing them with descriptions as “This can happen several times a day (e.g., TYT) or at weekly intervals (e.g., studies in the Corona Health app).” There are several other places with similar issue.

That’s a very good point! Abbreviations must be explained before they are used. We corrected this for every abbreviation used in the paper.

To provide some more study background info: The analyses happen with all apps on the so-called EMA questionnaires (synonym: assessment), i.e. the questionnaires that are filled out multiple times in all apps and the respective studies. This can happen several times a day (e.g., for the tinnitus study TrackYourTinnitus (TYT)) or at weekly intervals (e.g., studies in the Corona Health (CH) app). Nevertheless, the analysis happens on the recurring questionnaires, which collect symptoms over time and in the real environment through unforeseen notifications.

5. There are some typos. For example, on Page 3 last paragraph, in sentence “One can than calculate k performance scores and their standard deviation to...”, “than” should be changed to “then”.

Thank you spotting this typo. We corrected it and cross-checked the whole paper for more.

the validation fold-\cite{stone1974cross}. One can then calculate sk performance scores and their standard deviation to get an estimator for the performance of the model in the test set, which itself is an estimator for the model’s performance after deployment (see also Fig. \ref{fig:cross-validation-best-practice}). In addition, there exists the following strategies:

Reviewers' comments:

Reviewer #3 (Remarks to the Author):

The authors made great improvements on the paper. I appreciate the authors' efforts to provide detailed and thorough responses to my questions, and the newly added tables and figures that illustrate the studies and the terminologies. These are very helpful to readers. Yet I still have one feedback regarding to the pre-factor number 0.5 in function $f_{1}^{final} = f_{1}^{train} - 0.5\sigma(f_{1}^{train})$. If I understand correctly, the number 0.5 is just for illustration purposes. In detailed analysis, this number will be chosen accordingly. If that is the case, I would suggest using a Greek letter, say α , in the formula to show that this pre-factor is a hyperparameter, and explain its selection in the main context. Writing 0.5 directly in the formula may cause the confusion that readers should choose this number.

Point-by-Point-response #2

Reviewer #3 (Remarks to the Author):

The authors made great improvements on the paper. I appreciate the authors' efforts to provide detailed and thorough responses to my questions, and the newly added tables and figures that illustrate the studies and the terminologies. These are very helpful to readers. Yet I still have one feedback regarding to the pre-factor number 0.5 in function $f_1^{\text{final}} = f_1^{\text{test}} - 0.5\sigma(f_1^{\text{train}})$. If I understand correctly, the number 0.5 is just for illustration purposes. In detailed analysis, this number will be chosen accordingly. **If that is the case, I would suggest using a Greek letter, say α** , in the formula to show that this pre-factor is a hyperparameter, and explain its selection in the main context. Writing 0.5 directly in the formula may cause the confusion that readers should choose this number.

Thank you for your constructive feedback and for acknowledging the improvements made to the paper. We appreciate your suggestion regarding the pre-factor. Your interpretation is correct; the number 0.5 is indeed a placeholder for illustrative purposes. To address your concern and provide clarity to our readers, we will incorporate your suggestion and replace the number 0.5 with the α in the formula. Furthermore, we will expand on this change in the main context of the paper to explain the significance of α as a hyperparameter and the rationale for its selection.

Methods Section:

The hyperparameter α controls the importance of model robustness, with higher values of α emphasizing greater robustness.

`\label{fig:evaluation-schema}`
`\end{figure}`

We call one approach superior to another if the final score is higher. The final score to evaluate an approach is calculated as:

`\begin{equation}`
$$f_1^{\text{final}} = f_1^{\text{test}} - \alpha \sigma(f_1^{\text{train}})$$

`\end{equation}`

If the standard deviation between the folds during training is large, the final score is lower. The test set must not contain any selection bias against the underlying population. The pre-factor α of the standard deviation is another hyperparameter. The more important model robustness for the use case, the higher α should be set.

Results Section:

has the lowest final score. We recall the formula of the final score from the methods section: $f_1^{\text{final}} = f_1^{\text{test}} - 0.5 \sigma(f_1^{\text{train}})$. For these use cases, we set $\alpha = 0.5$. The greater the emphasis on model robustness and the increased concerns regarding concept drift, the greater the alpha value should be set.

REVIEWERS' COMMENTS:

Reviewer #3 (Remarks to the Author):

The authors have carefully addressed all my questions and I don't have any further comments. Thanks the authors for detailed explanations and revisions!